# Descriptions of the natural history of erythema nodosum leprosum to inform clinical classification – A semi-systematic review

Barbara de Barros [1]*, Saba Lambert[1], Vivianne Lopes Antonio Dias[2], Diana N.J. Lockwood[1], Stephen L. Walker[1]

**1** London School of Hygiene & Tropical Medicine, London, United Kingdom, **2** Universidade Federal Fluminense, Rio de Janeiro, Brazil

\* Barbara.de-barros@lshtm.ac.uk

## Abstract

### Background

Erythema nodosum leprosum (ENL) is a severe immunological complication of leprosy, characterised by painful nodules, fever, arthralgia, oedema, and systemic symptoms. Temporal classifications—acute, recurrent, and chronic—are inconsistently applied, complicating data comparisons. Standardised and agreed definitions are essential to ensure consistency in diagnosis, research, and clinical management.

### Objective

To examine how temporal classifications of ENL are used in modern literature and compare them to descriptions from the pre-corticosteroid era.

### Methods

We conducted a semi-systematic review of historical and contemporary literature. Historical texts published before 1940, prior to the introduction of sulfone antibiotics and corticosteroids, were purposively selected to capture descriptions of the natural history of ENL. For modern studies, we systematically searched PubMed, EMBASE, LILACS, SciELO, Scopus, African Index Medicus, Cochrane, and ClinicalTrials.gov from May 2024 to March 2025. The systematic review identified 572 articles after de-duplication, and 41 met inclusion criteria for providing definitions of ENL subtypes.

### Results

Five historical treatises were selected. Their clinical observations of nodular skin lesions with systemic symptoms—ranging in duration from weeks to months or even years—align with contemporary understandings of ENL. 41 studies included, at least one of the three temporal classifications (acute, recurrent, or chronic). The six-month criterion distinguishing acute and chronic ENL is used in all current definitions.

of the Creative Commons Attribution License, which permits unrestricted use, distribution, and reproduction in any medium, provided the original author and source are credited.

**Data availability statement:** All data underlying the results are available within the article and its supplementary file. This review synthesised information from published studies and historical sources, all of which are publicly available online. The translated historical text commissioned for this review is also provided in the Supporting Information.

**Funding:** This work was supported by The Hospital & Homes of St Giles grant ITCRZM25 to SLW. The funders had no role in study design, data collection and analysis, decision to publish, or preparation of the manuscript.

**Competing interests:** I have read the journal's policy and the authors of this manuscript have the following competing interested: BdB, SL, DNJL, and SLW are authors of some of the studies cited in this article and are members of the ENLIST group. The authors commissioned the English translation of Murata's article. These relationships have been disclosed in the interest of transparency and do not affect the integrity of the work.

However, definitions for recurrent and chronic ENL frequently overlapped, both referring to prolonged or multiple episodes after initial treatment, underscoring a lack of conceptual clarity.

## Conclusion

The absence of standardised ENL terminology impedes data comparison, meta-analysis, and clinical guideline development. A Delphi consensus process and longitudinal observational studies are recommended to refine, standardise agreed ENL classifications.

## Author summary

Erythema nodosum leprosum (ENL) is a painful and often chronic immune-mediated complication of leprosy. It causes red nodules on the skin with fever and other organ inflammation. ENL may be severe and challenging to treat usually requiring immunomodulatory drugs. Doctors and researchers often use the terms "acute," "recurrent," and "chronic" to describe different patterns of ENL, but these terms are not used consistently. This makes it difficult to compare findings of studies, optimise treatment and develop guidelines.

We selected five historical medical texts published before antimicrobial and immunosuppressive treatments were available to identify descriptions of ENL and its natural history. We then compared them with modern published definitions of ENL identified following a systematic search of eight databases.

Historical descriptions of ENL were largely consistent with what we see today, describing it as a chronic disease. We found general agreement about the duration of ENL for acute and chronic disease but the definitions of recurrent ENL overlapped with those of chronic ENL.

Our findings highlight the need for an agreed evidence-based classification of ENL. We recommend using clinical data and a consensus-building process, such as the Delphi method to develop standardised definitions. This would help improve the design of research studies for this challenging complication of leprosy.

## Introduction

Erythema nodosum leprosum (ENL, leprosy Type 2 reactions) is a severe, multisystem immune mediated complication of lepromatous leprosy (LL). ENL affects approximately 50% of individuals with LL and between 5% and 10% of those with borderline lepromatous (BL) leprosy [1,2]. A bacterial index of four or more is a risk factor for the development of ENL [3]. Leprosy is also complicated by Type 1 reactions which are

characterised by inflammation occurring in pre-existing skin lesions and nerves which may occur in all forms of leprosy but predominantly affecting individuals with the borderline forms [4].

ENL is associated with painful cutaneous and subcutaneous nodules, peripheral oedema, fever, nerve function impairment, arthritis, lymphadenitis, orchitis and, nasal and ocular involvement [3]. Inflammatory markers such as C-reactive protein are raised, and neutrophils are present in skin lesions [5]. The tender skin nodules occur in crops and may be present at different stages of evolution. Fever is a hallmark of the condition but may have settled by the time affected individuals are assessed [3].

ENL is a significant cause of morbidity and is associated with reduced health-related quality of life (HRQoL) compared to individuals without leprosy reactions [6–8]. ENL negatively affects people in many domains of their physical, psychological, social and economic lives [9,10]; and it is associated with catastrophic household costs associated with seeking-care and lost productivity.

The initial treatment for patients with ENL is high-dose oral corticosteroids [11,12]. Thalidomide is an effective alternative but is not available in many leprosy-endemic countries due to its teratogenicity [13]. Other agents including methotrexate, ciclosporin, apremilast and anti-tumour necrosis factor biological medications have been used [13–17].

In leprosy referral centres ENL is often a challenging condition to manage [18,19] requiring prolonged courses of corticosteroids which are associated with severe adverse effects such as hypertension, diabetes mellitus, severe infection, cataract and death [19–21]. Long-term administration of thalidomide for ENL is probably safer [21] but adverse effects may limit its use [21–23].

Effective, affordable and safe treatments for ENL are needed. An essential pre-requisite for developing evidence for effective treatments is ensuring that participants in research studies have the same condition and are similar. The ENLIST Group [24] proposed a descriptive definition of ENL for inclusion in the randomised controlled clinical trial of methotrexate and prednisolone [25]. For the purposes of the methotrexate trial, ENL was said to be present when "an individual with BL leprosy or LL develops 10 or more tender papular and/or nodular skin lesions".

The natural history of ENL is another important factor to consider in the design of robust research studies. Individual experiences of ENL may differ in severity, duration and the number of episodes. The ENLIST Group developed and validated the ENLIST ENL Severity Scale (EESS) to measure ENL severity [26]. However, there are no agreed definitions of classifying ENL with respect to chronicity and it is recognised that in many conditions there is a heterogeneous approach to classification according to duration [27].

ENL has been characterised as self-limiting condition, with episodes typically lasting from a few days to one or two weeks [28,29] and often resolving spontaneously [30,31]. This contrasts with cases in which individuals require treatment over several years [2,3,18,32,33]. Naafs has stated that three to four months treatment with prednisolone is too long because ENL episodes last less than one month "in most patients" [34] citing de Souza Araujo [35]. The WHO technical guidance on leprosy reactions recommends initiating treatment for severe ENL in adults with "moderate doses" of prednisolone (30–40 mg daily) without stating a duration but noting that recurrent and chronic ENL "require increased or prolonged doses…" [12]. The document is similarly vague about the duration of treatment with thalidomide as second-line treatment but indicates that high dose clofazimine may be needed for six months and a maintenance dose of 100 mg daily "for as long as ENL symptoms remain".

There are different ways of categorising the natural history of ENL. ENL may be short-lived, it may be episodic with or without periods of complete remission, or it may be persistent [2,3,32,36,37]. Individual ENL skin lesions may resolve whilst others arise during an individual episode. These factors may influence treatment responses, HRQoL and be potential confounders in clinical studies.

It is important that the classification of ENL is robust and accepted by the research community. Classification should be based on detailed clinical assessment, but it would be unethical to study the natural history of ENL by withholding treatment.

We summarised and critiqued the definitions of the natural history of ENL used in studies of ENL and identified data that may support their use or adaptation. We performed a semi-systematic literature review [38] combining a narrative review of ENL descriptions in the era prior to the use of effective anti-microbial therapy and corticosteroids with a structured search of databases for publications which include ENL classifications.

## Methods

This semi-systematic review was not prospectively registered, as it did not evaluate clinical outcomes or interventions but rather focused on historical and contemporary definitions and classifications of ENL.

### Narrative review

We purposively selected five texts of leprosy published before 1940 prior to the use of sulfone antimicrobial therapy or corticosteroids for leprosy and leprosy reactions respectively [39,40]. The rationale for choosing these texts was that two are considered the seminal descriptive texts of early biomedicine [41,42], one was the first to coin the phrase "erythema nodosum leprosum" (and subsequently considered an "excellent paper" despite not being widely cited) [43], one has been cited as evidence that ENL is of short duration [34,35] and one described ENL in the context of hydnocarpus oil treatment available prior to sulfone therapy [44].
The texts were reviewed for descriptions of ENL and its natural history.

### Systematic review

A systematic literature search was conducted between May 2024 and March 2025 in eight databases (PubMed (MEDLINE), EMBASE, LILACS, SciELO, Scopus, African Index Medicus, Cochrane and Clinicaltrials.gov). Keywords used were leprosy, Hansen's disease, hanseniasis, lepromatous leprosy, borderline lepromatous leprosy, multibacillary leprosy, erythema nodosum leprosum, ENL, type 2 reaction. Reference lists of included articles were reviewed, and international guidelines were checked. A search example is included in Box A in S1 Appendix.

### Inclusion criteria

We included clinical trials, observational studies, epidemiological cohorts, qualitative studies, case series and case reports which included a temporal definition of ENL. The search was not limited by date or language.

### Data management

Data were managed using Mendeley Reference Manager (Mendeley) for de-duplication of articles and screening. Microsoft Excel (Microsoft Corporation) was used for data extraction.

### Selection process

Initial screening of titles, abstracts and full text was conducted by the first author. Where there was uncertainty about the eligibility of an article a consensus was reached about inclusion with one of the other authors.

### Data collection process

The following data were extracted from each included publication: study design, aim, year of publication, setting, methodology, definitions of type of ENL and any citations for those definitions.

### Data synthesis

Descriptive statistics were used to summarise the characteristics of included studies. We conducted a narrative synthesis of ENL definitions, systematically comparing the terminology and definitions used in studies.

## Results

### Narrative review

We selected five texts (Table 1). The original versions of the texts were published in Norwegian (n = 2), Japanese, Portuguese and English. English and French language versions of the publications in Norwegian and the English version of Portuguese text were available. We used the original Portuguese text as two co-authors are native speakers and the French version as two of the co-authors are fluent French speakers. The Japanese text was translated by a professional translator with iterations reviewed by the authors and areas of uncertainty discussed and revised.

In 1847 prior to the discovery of *Mycobacterium leprae*, Danielssen and Boeck described episodes of fever followed by the sudden appearance of nodules accompanied by pain, malaise and lymphadenopathy [41]. They described the eruption as resembling erythema nodosum but made clear that it was not the same.

"This eruption closely resembles erythema nodosum, with which doctors, even skilled ones, have confused it."

In 1895, Hansen and Looft described individuals with "nodular leprosy" often developing nodules along with recurring episodes of fever, joint pain and general malaise, which could last from a few days to several months [42]. Hansen and Looft referenced the earlier work of Danielssen and Boeck and believed both descriptions were of the same phenomenon [42].

A modern reading of Hansen and Looft by Professor Magnus Vollset translated by him to English from the Norwegian text describes a clinical syndrome compatible with ENL [47]:

"This nodular form *[LL]* always advanced through outbreaks or eruptions accompanied by fever, repeating at longer or shorter intervals. For some the eruptions lasted only a few days with almost unnoticeable rise in body temperatures and almost no growth of the nodules. For others the fever could reach 40° C, last for weeks or months..."

**Table 1. Selected texts published prior to 1940 with descriptions of erythema nodosum leprosum.**

| Text | Language of version reviewed (original) | Description consistent with ENL | Time course of ENL described |
|---|---|---|---|
| Traité de la spédalskhed ou éléphantiasis des Grecs, par D. C Danielssen, William Boeck, traduit du norwégien, sous les yeux de M.D Danielssen par L.A Cosson, 1848 [41] | French (Norwegian [45]) | Fever, malaise, painful red nodules "Cette éruption ressemble assez à l'érythème noué (*erythema nodosum*) … " [*This eruption closely resembles erythema nodosum…*] | None |
| Leprosy: in its clinical and pathological aspects Hansen and Looft Translated by Norman Walker 1895 [42] | English (Norwegian) | "…suddenly a fresh outbreak of numerous nodules. The disease always advances by outbreaks of eruptions which repeated themselves at longer or shorter intervals… these outbreaks are always accompanied by fever…older nodules soften during a fresh outbreak, and completely or partly disappear…" | Few days to several years. |
| On Erythema Nodosum Leprosum Murata 1912 [37] | English* (Japanese) | "… erythema nodosum leprosum …the acute appearance of eruptions which cause significant suffering and often result in a critical worsening of patients' conditions." | 2 weeks to 12 months |
| A lepra: estudos realizados em 40 paizes [sic], de Souza-Araujo, 1929 [35] | Portuguese and English [46] (Portuguese and English) | "Lepra fever … is characterised by sudden eruptions accompanied by recurrent fever." | 4 weeks or more in 29.3% (all reaction types – see Table 2) |
| Leprosy Muir 1931 [44] | English (English) | "Previously noticeable leprous lesions swell up and become inflamed, and new lesions appear. There are general febrile symptoms…" | Few days or weeks to months or even years. |

*The English translation commissioned by the authors is available in S2 Appendix.

The term "erythema nodosum leprosum" was first introduced by Musuke Murata in 1912 [37], based on observations made between 1911 and 1912 at the Zensho Byoin in Higashimurayama, Tokyo, Japan. Murata examined 207 patients with nodular leprosy. Of these, 64 individuals (30.9%) developed ENL, 48 (75%) were men. The condition was described as an eruption of painful, inflamed skin lesions termed "netsukobu" (hot nodules) which were frequently accompanied by systemic symptoms including fever, headache, arthralgia, general malaise, palpitations, burning sensations, and loss of appetite.

Murata noted a higher frequency of ENL in individuals aged between 11 and 30 years of age. Symptoms duration varied considerably: in mild cases, regression occurred within two weeks, whilst moderate to severe cases took two to five months to resolve [37]. Murata wrote:

*"In 21 of the 64 cases [the eruptive period] lasted for 2 months; in 3 cases it lasted for 3 months; in 4 cases it lasted 5 months; and in one case alone did it persist for a year."*

Murata summarised the clinical course of ENL as symptomatic resolution taking three to four months. He documented that 13 (20.3%) individuals had further eruptive episodes, emphasising the recurrent nature of the condition. In a single case, episodes of eruptions continuously appeared and resolved throughout the year [37]. Murata described ENL as having "acute", "subacute", and "chronic" forms but did not define these terms.

Between July 1924 and January 1927, Heráclides César de Souza-Araújo, a Brazilian leprologist, visited leprosy services in 40 countries funded in part by the International Health Board of the Rockefeller Foundation and the Oswaldo Cruz Institute. In his report published in both Portuguese and English in 1929 de Souza-Araújo described his observations and interviews with leprosy and public health experts in each country. De Souza-Araújo spent a month at the Culion leprosy colony in the Philippines and described 11 forms of leprosy reactions ("lepra fever") delineated by the Culion leprologists [35], summarised in Table 2.

Several of the Culion classifications are consistent with ENL. Culion Type II most closely resembles ENL, with nodular, papular and vesicular eruptions. This pattern accounted for 42.2% of "lepra fever" cases at Culion. De Souza- Araújo

**Table 2. Culion Hospital leprosy reactions classification – adapted from de Souza-Araújo [35].**

| Culion Leprosy Reaction Classification | Description | Prevalence reported by de Souza Araújo n = 695 (%) | Interpretation of reaction using current terminology |
|---|---|---|---|
| Type I | Reaction of old lesions with or without fever | 107 (15.4) | Type 1 reaction |
| Type II | Eruptions of new lesions (maculopapular, papules, papulo-nodular, pustules, vesicles, etc) | 293 (42.2) | ENL |
| Type III | Eruption of new lesions and reaction of old lesions | 83 (11.9) | Type 1 reaction and/or ENL |
| Type IV | Successive reactions separated by short intervals | 86 (12.4) | ENL |
| Type V | Fever attacks with delayed skin lesions | 5 (0.7) | ENL |
| Type VI | Prolonged and severe fever attacks without skin lesions | 3 (0.4) | |
| Type VII | Neuritis and neuralgic pain, with or without noticeable neuritis | 24 (3.4) | Neuritis |
| Type VIII | Rheumatoid, articular and muscular manifestations associated (with "lepra fever reaction") | 24 (3.4) | Possibly ENL |
| Type IX | Iritis, conjunctivitis, or other acute ocular inflammation probably related to leprosy | 62 (8.9) | Possibly ENL |
| Type X | Orchitis | 8 (1.1) | ENL |
| Type XI | Generalised lymphadenitis accompanied by fever and malaise | Added a posteriori | Possibly ENL |

states that 29.3% of "lepra fever" lasted for 4 weeks or longer, however, later in the chapter in the Portuguese version he refers to the same proportion lasting 4 weeks, but this discrepancy does not occur in the English version which refers to "4 [weeks] or more" and "four or more weeks" [35,46]. The passages below are quoted verbatim from the original Portuguese and English texts:

*"Quanto á [sic] duração da reacção [sic], foi de 2 semanas em 32,4%, de 4 ou mais em 29,3%, de uma em 18,7% e de 3 semanas em 18,6%."* [35] [Portuguese version]

*"The duration of the reaction was of two weeks in 32.4%; of 4 or more [weeks], in 29.3%; of one [week], in 18.7%; and of 3 weeks, in 18.6 per cent."* [46] [English version]

*"A duração foi de uma semana em 18,7%; de 2 semanas em 32, ou mais em 29,3%, de uma em 18,7% e de 3 semanas em 18,6% e de 4 semanas em 29,3%."* [35] [Portuguese version]

*"Duration: 18.7% one week; 32.4% two weeks; 18.6% three weeks and 29.3% four or more weeks."* [46] [English version]

Culion Types III, IV and V are consistent with ENL and Types VIII, IX, X and XI consistent with extra-cutaneous organ involvement in ENL.

Of the five texts reviewed, four provided some description of the time course of ENL ranging from several weeks to years.

Ernest Muir, in 1931, described leprosy reactions both as a complication of treatment and a presenting complaint. With respect to the duration of leprosy reactions Muir stated that:

*"Lepra reaction may disappear again within a few days or weeks, or it may linger on for months or even years."* [44]

## Systematic review

The search resulted in 572 records after de-duplication (Fig 1). Reviewing the references resulted in one additional publication. The oldest article screened was published in 1956. One study from 1981 could not be obtained [48]. The list of all publications included in this review, along with their characteristics, is provided in Table A in S1 Appendix.

A total of 153 publications were assessed, and 41 contained definitions of ENL classifications (Table C in S1 Appendix). One hundred and eleven fully screened texts classified ENL but did not provide definitions for the ENL categories. The 41 publications included case reports, systematic and other literature reviews, technical guidance, cross-sectional, prospective and retrospective cohorts, clinical trial registrations and clinical trials. The included studies were published between 2003 and 2025. Fifteen (36.6%) of the studies were conducted in India.

All 41 publications used at least one of the terms "acute" or "recurrent" or "chronic" to classify ENL. In four (9.7%) studies, other terminology was used to define ENL, including "recurrent multiple", "acute multiple" and "repeat ENL episode". Twenty-six (63.4%) studies defined acute ENL, 39 (95.1%) defined chronic ENL, and 29 (70.7%) defined recurrent ENL. Of the 41 publications with defined ENL classifications, 23 (56.1%) provided a citation for the definitions.

## Acute ENL

Twenty-six publications defined acute ENL using four definitions which are summarised in Table 3.

The first definition of acute ENL was published in 2006 by Pocaterra *et al.* [2], who described a retrospective cohort of 481 individuals with BL leprosy and LL in Hyderabad, India. 116 (24.1%) were diagnosed with ENL. Acute ENL which affected only 5 (4.3%) of the cohort was defined as ENL lasting less than six months and not requiring an increase in prednisolone dose.

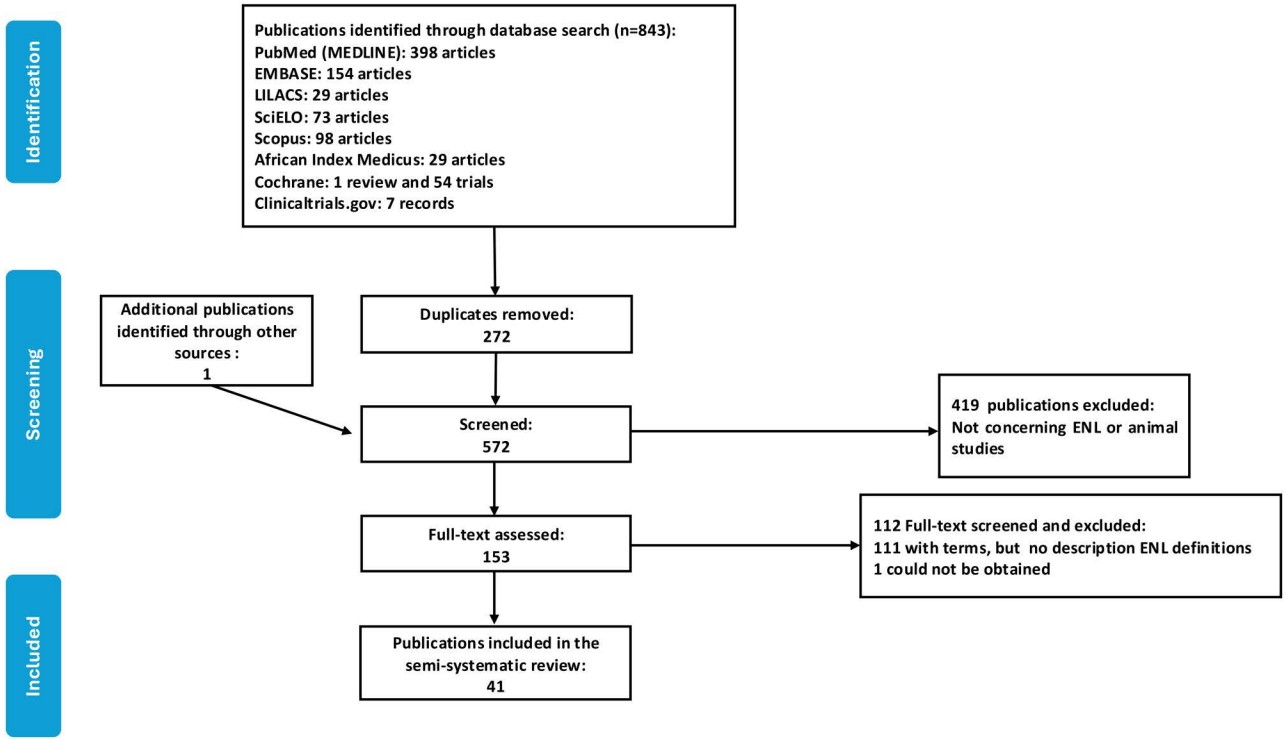

**Fig 1. Flow diagram of included studies.**

**Table 3. Definitions of acute ENL in included studies with the number of subsequent citations.**

| Author, year | Definition | Number of subsequent citations of the definition |
|---|---|---|
| Pocaterra, 2006 [2] | "…one ENL episode lasting less than six months with a steady decrease in steroid tapering, no recurrence of ENL when receiving prednisolone, and no increase in severity requiring an increased steroid dose." | 5 |
| Walker, 2014 [20] | "… a single episode [of ENL] lasting less than 24 weeks." | 16 |
| de Barros, 2020 [25] | "… a first episode of ENL less than 24 weeks in duration or a second or subsequent ENL episode, lasting less than 24 weeks and occurring 84 days (i.e., 12 weeks) or more after stopping treatment for ENL." | 1 |
| Baima de Melo,2020 [49] | "… if it was the first episode [of ENL] or if the last episode had occurred more than 24 weeks before and the patient had not been under treatment in that interval; …" | 0 |

Walker *et al.* defined acute ENL as a single episode lasting less than 24 weeks irrespective of treatment status as part of a retrospective study of 414 patients with leprosy in Ethiopia, 99 (23.9%) had ENL of whom 19 (19.2%) met the definition of acute ENL [20].

The two later definitions of de Barros [25] and Baima de Melo [49] extended the definition of acute ENL to subsequent episodes occurring more than 12 weeks and 24 weeks respectively after cessation of ENL treatment.

## Recurrent ENL

Definitions of recurrent ENL were heterogeneous and are summarised in Table 4.

Kumar *et al*. published a retrospective study of leprosy reactions in Chandigarh, India which included the first definition of recurrent ENL [32]. Of 1494 multibacillary patients, 337 (22.5%) experienced ENL. Recurrent reactions (both Type 1 reactions and ENL) were defined as the return of symptoms more than six weeks after completion of treatment for reactions. In this cohort, 217 (64.4%) individuals with ENL experienced more than one "episode" of ENL. If symptoms returned within 6 weeks, this was considered due to inadequate or abruptly discontinued treatment.

Walker *et al*. [20] reported 10.1% had recurrent ENL in the Ethiopian cohort using their definition. Kar *et al*.[50] conducted a prospective single-centre study including 40 individuals defined as having recurrent or chronic ENL, recruited according to their classification.

Mishra *et al*.[51] performed a prospective observational study that involved 30 individuals in two treatment groups and reported ENL recurrence rates of 20% in one group and 34% in the other.

The proposed minimum durations between episodes of ENL ranged from 15 days to "3 months" to distinguish between recurrent and chronic episodes. Only de Barros defined an upper limit (84 days) to distinguish recurrent from a subsequent episode of acute ENL. Mishra defined recurrent ENL as four to seven episodes per year. None of the studies proposing definitions for recurrent ENL in Table 4 provided clinical data to support the choice of interval between episodes of ENL.

## Chronic ENL

Table 5 summarises the definitions of chronic ENL. In a review article of the use of thalidomide Okafor [52] reported that Sheskin and Convit [53] had enrolled participants with "chronic ENL" into a trial of thalidomide. Okafor defined chronic ENL as more than three months which is the earliest definition of chronic ENL we identified. Sheskin and Convit enrolled

**Table 4. Definitions of recurrent ENL in included studies with the number of subsequent citations.**

| Author, year | Definition | Number of subsequent citations of the definition |
|---|---|---|
| Kumar, 2004 [32] | "… a recurrence of symptoms [of ENL] … more than six weeks after completion of treatment for reaction." | 1 |
| Walker, 2014 [20] | "… a second or subsequent episode of ENL occurring 28 days or more after stopping treatment for ENL …" | 23 |
| Kar, 2016 [50] | "… [ENL] which relapsed after 3 months of stopping anti-reactional treatment." | 0 |
| de Barros, 2020 [25] | "… a second or subsequent episode of ENL occurring between 28 and 84 days after stopping treatment for ENL." | 0 |
| Mishra, 2022 [51] | "… four to seven episodes [of ENL] per year, with or without treatment." | 0 |

**Table 5. Definitions of chronic ENL in included studies with the number of subsequent citations.**

| Author, year | Definition | Number of subsequent citations definition |
|---|---|---|
| Okafor, 2003 [52] | "…chronic ENL (> 3 mo *[months]*)." | 0 |
| Kumar, 2004 [32] | "… continued antireaction treatment for a period of 6 months or more." | 16 |
| Walker, 2014 [20] | "… [ENL] occurring for 24 weeks or more during which a patient has required ENL treatment either continuously or where any treatment free period has been 27 days or less." | 20 |
| Kar, 2016 [50] | "… [ENL] reactional episodes which persist more than 6 months or shows relapse of reaction within 3 months of stopping anti-reactional treatment." | 0 |

participants with "lepra reaction" and this undoubtedly included individuals with ENL. However, they did not define ENL duration, nor did they use the term "chronic" but reported that the minimum duration of lepra reaction in participants was three months [53].

Kumar *et al.*[32] defined chronic ENL as requiring continuous treatment for 6 months or longer. In keeping with Kumar others have defined chronic ENL as disease activity requiring treatment (and/or persisting) for 24 weeks or longer. Additionally, Walker *et al.* [20] stipulated that treatment free periods were less than 28 days and 70.7% of individuals with ENL fulfilled these criteria for chronic ENL. Kar *et al.* [50] defined treatment free periods as less than three months for chronic ENL.

### Other classifications of ENL

Other temporal classifications of ENL appeared in the reviewed publications as summarised in Table 6.

Pocaterra used the term "acute multiple ENL" interchangeably with "acute recurrent" episodes. These terms appear infrequently in the literature possibly resulting from lack of precision and potential redundancy within the classifications of ENL.

### Discussion

The historical descriptions of leprosy and leprosy reactions provide context for the classification of ENL. The clinical observations described in the five monographs of nodular skin lesions associated with systemic illness of variable severity are consistent with the clinical signs categorised as ENL today. The episodic nature of ENL is also a feature described by authors from the era before the availability of effective anti-microbial and immunomodulatory agents [35,37,42]. The length of these episodes varied significantly. Some were brief, lasting only a few days with minimal systemic symptoms. Others were more prolonged, continuing for weeks, months or even years. Some individuals are described as experiencing "multiple eruptions" each year for several years [42]. Murata classified ENL as being "acute", "subacute", and "chronic" but neither he nor the authors of the other four texts selected reported the interval between episodes. An "episode" of ENL implied an inflammatory event marked by spontaneous symptom resolution. However, this clarity is lost in those whose symptoms are controlled with immunosuppressive therapy [55–57].

The six-month criterion distinguishing acute and chronic ENL is used in all current definitions. However, definitions of recurrent and chronic ENL overlap substantially, with both categories referring to multiple or prolonged episodes following initial treatment.

The definitions of recurrent ENL are based on the length of treatment free intervals between episodes which range from 15 days to 84 days (or 3 months) except for Mishra *et al.* [51] who defined recurrent ENL as occurrence of four to seven episodes per year, with or without treatment. Mishra's definition introduces some overlap with chronic ENL as it includes individuals on treatment for ENL.

Table 6. Definitions of other temporal classifications of ENL in included studies with the number of subsequent citations.

| Author, year | Classification | Definitions | Number of subsequent citations |
|---|---|---|---|
| Pocaterra, 2006 [2] | Acute recurrent ENL | "… multiple discrete episodes [of ENL]; …" | 1 |
| Pocaterra, 2006 [2] | Acute multiple ENL | "… more than one ENL episodes with the same characteristics as acute single ENL." | 2 |
| Maghanoy, 2017 [54] | Repeat ENL episode | "...a new crop of ENL lesions that was detected by the physician at least 4 weeks after old ENL lesions were subsided and/or 4 weeks after steroid treatment was discontinued." | 0 |

Chronic ENL is defined by most as exhibiting continuous or near-continuous symptoms requiring treatment for more than 6 months. Two definitions allowed treatment-free interval. Walker *et al.*[20] used a period without treatment of 27 days or fewer to define the same episode of chronic ENL whereas Kar *et al.*[50] defined the treatment-free period as less than 3 months.

The cross-sectional ENLIST 1 study enrolled 292 participants at seven leprosy referral centres and reported that there was an almost even distribution of ENL types: acute 34.2%, recurrent 32.5%, and chronic 33.2% [3]. Baima de Melo *et al.* [49] using different definitions reported rates in 26 individuals with "active" ENL in Brazil of acute ENL 15.4%, recurrent 26.9% and chronic 57.7%. This difference may be explained by the different time of clinical evaluation or different definitions. Kar *et al.* observed that 24% of the participants initially classified as having acute ENL experienced subsequent episodes, suggesting progression to recurrent or chronic forms [50]. Studies from leprosy referral centres using longitudinal data (albeit with different ENL definitions) report rates of acute ENL of 4.3-19.9%, recurrent ENL 10–77%, and chronic ENL 18–70% [2,18–20,32]. This difference between cross-sectional and retrospective studies with longitudinal data may indicate a progression from acute to recurrent or chronic ENL. These data need to be interpreted with caution as people with chronic or severe ENL are likely to continue to attend leprosy referral centres for treatment.

The term "episode" which occurs in multiple definitions is not clearly defined. Pocaterra *et al*. described the end of an ENL episode when "steroid tapering ended". Walker [20] defined an ENL episode as the occurrence requiring the institution or change of treatment; however, de Barros subsequently defined "flare" as a symptomatic exacerbation of ENL [25]. The overlap between the terms "episode" and "flare" highlights continued ambiguity in ENL terminology.

We propose that the term episode should refer to the period from the onset of ENL symptoms until the individual is free of ENL symptoms. The duration of the interval between episodes should determine whether an individual has acute or recurrent or chronic ENL. The term "flare" should be reserved for a deterioration of ENL whilst an individual is taking treatment for ENL.

The recurrence of ENL following remission, or deterioration as treatment is reduced ("flare") is a particularly challenging aspect of the condition. High rates of "recurrence" and/or "flaring" of ENL are reported. In India, 64.3% of individuals had "recurrent" episodes [32] as did a similar proportion in Ethiopia (63%) [58]. Kumar *et al.* argued that recurrence of a reactional episode within 6 weeks is due to inadequate anti-reaction treatment [32] it is unclear how one could differentiate between sub-optimal treatment of recurrent ENL from chronic ENL. The distinction between the onset of a new episode of ENL and deterioration of ENL whilst on treatment may have useful prognostic value. It is reasonable to hypothesise that ENL which deteriorates on immunomodulatory treatment may behave in a clinically different manner to ENL recurring following a treatment free period.

The classification of leprosy reactions has often been challenging for leprologists. In 1949 de Souza Lima [59] expressed dissatisfaction with the prevailing classification of leprosy reactions and highlighted the need to establish clear definitions agreed by the international leprosy community. This sentiment was echoed by Pettit and Waters [60], who shared the view that ENL represented an acute exacerbation of the chronic disease of leprosy.

In the early period of sulfone use for leprosy treatment, some researchers hypothesised that ENL was an adverse effect of therapy [61]. Others interpreted ENL as an indicator of a favourable response [62]. However, as Pettit and Waters highlighted and we have shown, the literature prior to the introduction of sulfone anti-microbial therapy is replete with descriptions of ENL [60]. The nodular form of leprosy described by Hansen and Looft was complicated by ENL, and the affected individual eventually succumbed. As Vollset describes in his translation from Hansen and Looft:

> "Intermittently all affections [fever and nodules] could spontaneously disappear and the patient would heal, but in general the life of the patient would end after eight or nine years" [47]

In the WHO multi-drug therapy (MDT) era, ENL occurs before, during, or after the completion of treatment [13]. It is hypothesised that bactericidal action of MDT in individuals with high bacillary load results in *Mycobacterium leprae* antigens inducing immune activation leading to an inflammatory response [63]. The pathophysiology of ENL is not well

understood, there is evidence that pro-inflammatory cytokines such as tumour necrosis factor alfa, interferon gamma and interleukins 1 and 6 may be increased [5], all of which may contribute to ENL severity [64].

It is unclear whether immunosuppressive therapy influences the duration or type of ENL. Individuals may require corticosteroids for prolonged periods and some experience tachyphylaxis [12,22]. The reported duration of ENL "episodes" (in individuals prescribed ENL treatment) ranges from 3 to 72 days [2,3,55]. However, this may not reflect the length of exposure to high dose corticosteroids for those who have multiple episodes. The prolonged duration of treatment for chronic ENL is associated with morbidity and mortality [20,21].

Corticosteroids are the recommended first-choice treatment for patients with ENL [12]. The influence of treatment on the natural history of ENL is uncertain. Corticosteroids are not considered disease-modifying agents in other inflammatory conditions such as rheumatoid arthritis [65,66]. Whilst they provide rapid symptomatic relief, their role in altering the underlying disease trajectory is limited and generally reserved for short-term "bridging" therapy [65]. Identifying a disease modifying agent for ENL that would shorten the period individuals experience symptoms would be a significant advance in management.

We have identified definitions of the natural history of ENL and their relative frequencies in the literature. We have identified weaknesses in all current definitions and suggested improvements. The current lack of standardised terminology hinders data comparison, meta- analyses, and clinical guideline development. To address the current inconsistencies, the Delphi technique [67] offers a structured method for achieving expert consensus. Observational prospective studies and analysis of routine data of individuals who decline ENL treatment would enable validation and refinement of proposed definitions, ensuring they are both evidence-based and clinically meaningful. A clear standardised clinical ENL classification is vital to improving our understanding of ENL and improving the lives of those affected by this condition.

## Limitations

The limitations of this study are reliance on translated texts in our narrative review. The definitions of the types of ENL in the reviewed studies were often reported inconsistently in the methods and sometimes presented in the results or discussion making analysis challenging.

## Supporting information

**S1 Appendix. This file contains the Table A with online resources used in the narrative review, Box A with the MEDLINE (Ovid) search strategy, Table B with the full list of included studies used in this semi-systematic review and Table C includes extracted study characteristics and definitions of ENL.**
(PDF)

**S2 Appendix. English translation On Erythema Nodosum Leprosum file: This document contains the translation commissioned by the authors for the purposes of this semi-systematic review.**
(PDF)

**S3 Appendix: PRISMA 2020 checklist.** PRISMA 2020 checklist adapted from Page MJ et al. *The PRISMA 2020 statement: an updated guideline for reporting systematic reviews.* BMJ. 2021;372:n71. https://doi.org/10.1136/bmj.n71. Licensed under CC-BY 4.0.
(PDF)

## Acknowledgments

We would like to express our gratitude to the people with lived experience of leprosy who have shaped our understanding of the condition. We acknowledge that some information used in this work was obtained through observations of people affected by leprosy who may not have been allowed agency to object to being included in the work in which their information was included. We believe their important contribution should be acknowledged.

We would like to thank the librarians and information technologists at the London School of Hygiene & Tropical Medicine, London, UK, FIOCRUZ, Rio de Janeiro, Brazil and in particular Teruko Sekiguchi, Embassy of Japan in London, UK. Dr Églantine Lebas participated in discussions of the French translation of Danielssen and Boeck.

We would like to acknowledge the other members of the ENLIST Group:

Bishwanath Acharya, Medhi Denisa Alinda, Bhagyashree Bhame, Marivic Balagon, C. Ruth Butlin, Joydeepa Darlong, Shimelis N. Doni, Farha Sultana, Alemtsehay Getachew, Deanna A. Hagge, Abdulnaser Hamza, M. Yulianto Listiawan, Milton O. Moraes *(in memoriam)*, Neeta Maximus, Indra Napit, Jose da Costa Nery, Peter Nicholls, Vivek V. Pai, Anna Sales, Mahesh Shah, Anju Wakade

## Author contributions

**Conceptualization:** Barbara de Barros, Stephen L. Walker.

**Data curation:** Barbara de Barros.

**Formal analysis:** Barbara de Barros.

**Investigation:** Barbara de Barros, Vivianne Lopes Antonio Dias.

**Methodology:** Barbara de Barros, Stephen L. Walker.

**Resources:** Vivianne Lopes Antonio Dias.

**Supervision:** Diana N.J. Lockwood, Stephen L. Walker.

**Writing – original draft:** Barbara de Barros.

**Writing – review & editing:** Barbara de Barros, Saba Lambert, Vivianne Lopes Antonio Dias, Diana N.J. Lockwood, Stephen L. Walker.

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
