## [Decision Letter · Decision Letter 0]

5 Jan 2026

PNTD-D-25-01473

Descriptions of the natural history of erythema nodosum leprosum to inform clinical classification – a semi-systematic review

Dear Dr. de Barros,

Thank you for submitting your manuscript to PLOS Neglected Tropical Diseases. After careful consideration, we feel that it has merit but does not fully meet PLOS Neglected Tropical Diseases's publication criteria as it currently stands. Therefore, we invite you to submit a revised version of the manuscript that addresses the points raised during the review process.

Please submit your revised manuscript within by Mar 06 2026 11:59PM. If you will need more time than this to complete your revisions, please reply to this message or contact the journal office at plosntds@plos.org. Please include the following items when submitting your revised manuscript:

We look forward to receiving your revised manuscript.

Kind regards,

Elsio A Wunder Jr, DVM, Ph.D.

Section Editor

Elsio Wunder Jr

Section Editor

Shaden Kamhawi

co-Editor-in-Chief

Paul Brindley

co-Editor-in-Chief

**Additional Editor Comments (if provided):**

**Journal Requirements:**

**Reviewers' Comments:**

Reviewer's Responses to Questions

**Key Review Criteria Required for Acceptance?**

**Methods**

-Are the objectives of the study clearly articulated with a clear testable hypothesis stated?

-Is the study design appropriate to address the stated objectives?

-Is the population clearly described and appropriate for the hypothesis being tested?

-Is the sample size sufficient to ensure adequate power to address the hypothesis being tested?

-Were correct statistical analysis used to support conclusions?

-Are there concerns about ethical or regulatory requirements being met?

Reviewer #1: (No Response)

Reviewer #2: The objectives of the study are clearly articulated and focus on examining how temporal classifications of erythema nodosum leprosum (ENL) have been applied in contemporary literature, as well as comparing these classifications with descriptions from the pre-corticosteroid and pre-antibiotic era. While a formal testable hypothesis is not explicitly stated, the aims are clearly defined and appropriate for a review-based study.

The study design is appropriate to address the stated objectives. The combination of a narrative review of historical texts with a systematic review of contemporary literature is suitable for exploring conceptual and classification-related questions, rather than testing causal associations.

As this work is a review study, no study population in the traditional sense is included. Instead, the selection of historical texts and contemporary publications constitutes the analytical corpus. The sources included are generally appropriate for the objectives of the review, although clearer justification of the selection criteria would further strengthen the methodology.

Similarly, considerations of sample size and statistical power are not directly applicable, as this is not a primary quantitative study. The conclusions are based on qualitative synthesis of historical and contemporary literature rather than statistical inference.

Formal statistical analyses were not required for the primary objectives of the study, and the conclusions are supported by descriptive and comparative assessment of the included literature. Where numerical data are presented, they are used descriptively and appropriately.

There are no apparent concerns regarding ethical or regulatory requirements. The study is based exclusively on previously published literature and historical texts and does not involve human participants, patient data, or identifiable personal information.

Reviewer #3: Most of the questions are not relevant as the manuscript is a semi systematic review. However the methods used are appropriate for the purpose of the study.

**Results**

-Does the analysis presented match the analysis plan?

-Are the results clearly and completely presented?

-Are the figures (Tables, Images) of sufficient quality for clarity?

Reviewer #1: (No Response)

Reviewer #2: The analyses presented are generally consistent with the stated methodological approach.

Overall, the results are clearly presented and cover the main objectives of the study. That said, certain sections would benefit from improved clarity and conciseness, particularly where repetitive citations or overlapping descriptions affect readability. Explicit clarification of whether specific findings derive from historical sources or contemporary studies would further enhance completeness and transparency.

The figures and tables are generally of adequate quality and contribute meaningfully to the presentation of the results. Nonetheless, some tables could be simplified to avoid redundancy with the accompanying text, and greater consistency in table titles and structure would improve clarity and facilitate comparison across sections.

Reviewer #3: the results are clearly stated and match the objectives of the systematic review.

**Conclusions**

-Are the conclusions supported by the data presented?

-Are the limitations of analysis clearly described?

-Do the authors discuss how these data can be helpful to advance our understanding of the topic under study?

-Is public health relevance addressed?

Reviewer #1: (No Response)

Reviewer #2: The conclusions are generally supported by the data presented, particularly with respect to the identification of inconsistencies and overlaps among current temporal classifications of erythema nodosum leprosum. The historical and contemporary sources cited provide adequate support for the authors’ central arguments, although clearer articulation of the proposed implications for classification would further strengthen the conclusions.

The authors do discuss how the data contribute to advancing understanding of the topic, especially by highlighting conceptual weaknesses in existing ENL classifications and emphasizing the need for harmonization. Further elaboration on how these findings could inform future research frameworks or clinical classification systems would add value.

The public health relevance of the study is implicit rather than explicit. While the implications for clinical management and research comparability are evident, a more direct discussion of how improved ENL classification could impact patient care, treatment strategies, and health policy, particularly in endemic settings, would strengthen this aspect of the manuscript.

Reviewer #3: the conclusions and the recommendations are supported by the results presented. Since ENL is a chronic, debilitating condition, contributing the burden of leprosy worldwide, the recommendation that a consensus should be reached in the classification of ENL is a valid one.

**Editorial and Data Presentation Modifications?**

Reviewer #1: (No Response)

Reviewer #2: Please find below minor suggestions, organized by section, intended to contribute to the improvement of the manuscript.

KEY-WORDS (line 9): The keyword “leprosy” appears to be missing and should be included.

ABSTRACT

Methods

Line 24: The historical texts appear to have been included based on their significance rather than as a result of the systematic review process. We suggest clarifying this distinction.

Results

Line 33: For greater clarity, it would be helpful to specify that the 41 studies refer to the contemporary literature, in contrast to the historical treatises mentioned in the preceding sentence.

Author Summary

Lines 55–56: It appears that other historical medical texts were available, but the five included were selected due to their importance. This selection criterion should be made explicit.

Line 65-66: Please review the sentence and replace one instance of the word “improve” to avoid repetition and enhance clarity.

INTRODUCTION

Lines 114-115: Please review the sentence and replace or remove one instance of the word “conditions” to improve clarity.

METHODS

Narrative review

Lines 147 148: Please provide a clearer explanation of the rationale for including texts published prior to 1940, addressing both their historical relevance and the fact that the resolution of ENL symptoms may be influenced by antibiotic and immunosuppressive therapies, which were introduced after that period. At this stage of the manuscript, this selection criterion is not clear to the reader. The “inclusion criteria” later defined in the Methods section as a “temporal definition of ENL” does not appear to be appropriate for this selection.

The sentence included in the Results section (lines 206–207), “The rationale for choosing these texts was that two are considered the seminal descriptive texts of early biomedicine,” would be more appropriately presented as a methodological justification for this selection. As such, it appears better suited for inclusion in the Methods section rather than in the Results.

Systematic review

Lines 155-161: Considering the high specificity of the leprosy literature and the existence of two major international journals dedicated to this disease—Leprosy Review (first available issue in 1928) and the International Journal of Leprosy and Other Mycobacterial Diseases (first available issue in 1933), both of which provide advanced search tools (http://leprev.ilsl.br/arquivo.php and http://ijl.ilsl.br/edicoes_anteriores.php) we recommend that these sources also be consulted to minimize the risk of missing relevant and important publications.

These sources could also be used to identify additional historical publications from the period preceding the introduction of antibiotics and corticosteroids for the treatment of leprosy reactions.

Lines 167-194: It should be noted that the sections included in this part of the manuscript (inclusion criteria, data management, selection process, and data collection process) appear to relate primarily to the systematic review rather than to the narrative review. For the sake of clarity and to facilitate reader understanding, it would be advisable to subdivide the Methods section into two clearly defined components: (1) Narrative Review and (2) Systematic Review. The methodological elements of the systematic review could then be presented as subsections (e.g., 2.1 Inclusion criteria, 2.2 Data management, etc.). Alternatively, the authors may consider an alternative structure that more clearly delineates these distinct methodological approaches.

RESULTS

Table 1: Considering that not all texts published prior to 1940 were included, we suggest that the title of this table be revised to use “Selected texts.”

The third column (The description consistent with erythema nodosum leprosum) appears to be repetitive of the text that follows the table. The authors may wish to consider restricting the content of the table to the title, authors, year of publication, language, and the time course described, in order to improve the fluency and readability of the text.

Lines 229-236: In the Methods section, it is stated that the original French version of Hansen and Looft was used, as two of the co-authors are fluent French speakers. In this context, it is unclear why the same manuscript is also cited using the English translation by Professor Magnus Vollset. Was the clinical description compatible with ENL not already adequately presented in the original French version?

Furthermore, if the intention was to support the conclusion (possibly from Vollset) that the nodular form corresponds to the contemporary classification of lepromatous (LL) leprosy, we would recommend avoiding this approach, considering that such a conclusion cannot be reliably drawn from the historical descriptions alone, as borderline-lepromatous cases may present a similar clinical appearance and may also develop ENL.

Table 2: In the first row (Type I), it is intriguing to note the description of fever, which would be more consistent with ENL. In the third row (Type III reaction), the description of a “reaction of old lesions” is more likely to correspond to a reversal reaction rather than ENL, although the authors may be referring to the worsening of old nodular lesions, corresponding to reactivation of subcutaneous nodules from previous episodes of ENL.

Therefore, in both instances, we would suggest that the authors exercise the same level of caution applied to the classification of Types VIII and IX in the Cullion classification and consider using the wording “Type 1 reaction and possibly some cases of ENL” for rows 1 and 3 (Types I and III).

Lines 276-286: Repetitive citations from original texts hinder readability. Therefore, we recommend retaining only those excerpts that present divergent translations.

Systematic review

Lines 301-304: As suggested in the comments about the Methods section for the systematic review, a simple search of abstracts using only the term “erythema nodosum leprosum” resulted in 5 publications prior to 1956, which may or may not cope with your inclusion criteria, but we suggest checking them:

- Volume: 19 - Edition: 3 - Jul/Aug/Sep - 1951

"Erythema nodosum" and "Erythema multiforme"

- Volume: 23 - Edition: 1 - Jan/Feb/Mar - 1955

The histopathology of acute panniculitis nodosa leprosa (erythema nodosum leprosum)

- Volume: 15 - Edition: 4 - Oct/Nov/Dec - 1947

Erythema nodosum in leprosy

- Volume: 14 - Edition: 1 - Jan/Dec - 1946

Erythema nodosum in leprosy. A study of the pathogenesis with reference to carbohydrate metabolism

- Volume: 5 - Edition: 4 - Oct/Nov/Dec - 1937

Erythema nodosum leproticum

Line 328: It would be preferable if the title of Table 3 followed the same format as those of Tables 4 and 5.

Line 331-333: Please review the statement that 116 cases (24.1%) “were diagnosed with ENL,” as it is unclear whether erythema nodosum leprosum was present at the time of leprosy diagnosis or whether these patients were diagnosed with ENL during the course of the cohort study.

Line 366: It is noteworthy that this author (Mishra) used definitions that are peculiar and markedly different from those employed in all other studies. We believe that this discrepancy, and its comparison with other studies, should be highlighted in your Results section.

Table 6: If available, please expand the definition of acute recurrent ENL proposed by Pocaterra (2006) to facilitate a clearer understanding (first row).

DISCUSSION

First paragraph (lines 401–412): As the main objective was to compare temporal classifications of ENL used in modern literature them with descriptions from the pre-corticosteroid era, it is important to recognize that no historical temporal classification of ENL was identified. Although the authors identified a historical classification used at Culion Hospital and adapted from de Souza-Araújo, this classification was based on clinical features only and did not include any temporal criteria.

Line 410: We suggest including the word “spontaneous” before “symptoms resolution.”

Lines 450-453: Does acute ENL imply the presence of any interval between two or more episodes, given that it is defined as lasting less than six months? It is important to consider whether a patient with ENL could experience an initial episode, followed by regression, recurrence after the expected interval of 1–3 months, and subsequent regression without further flares, all within a total period of less than six months.

On this basis, it would be reasonable to interpret acute ENL as referring to a patient with a single episode of ENL lasting less than six months. If this rationale is accepted, no intervals between episodes would be expected. In this context, references to an “interval” would be appropriate only for recurrent or chronic ENL.

Lines 472-473: We believe that the hypothesis that ENL could represent an adverse effect of therapy should not merely be mentioned in the Discussion but rather dismissed on the basis of your findings from the pre-sulfone descriptions.

General observations for the Discussion section

In the Results section, it is stated that the first definition of acute ENL was published in 2006 by Pocaterra et al. (line 331); however, there is no indication of when the definitions of recurrent and chronic ENL were introduced. If this information was identified during your review, we believe it would be valuable to report and cite it.

Lines 415-416: We suggest that this overlap, and the resulting difficulty in classifying certain patients, be explicitly acknowledged, as this represents a key conclusion of the manuscript. Highlighting this issue would strengthen the Discussion and support ongoing efforts to refine and harmonize ENL classification. Accordingly, the discussion of this topic could be expanded to remind readers of this overlap, noting that the definition of recurrent ENL includes periods without medication between episodes that may be as short as two weeks, whereas chronic ENL is described as having “near-continuous symptoms.”

Lines 477-482: The notion of short survival among patients as a consequence of ENL should be interpreted with caution. According to Irgens et al. (Irgens LM, Nedrebø Y, Sandmo S, Skvennes A. Leprosy. Bergen, Norway: Leprosy Archives of Bergen; 2006), 86% of the 754 patients admitted to St. Jørgens Hospital in Bergen between 1822 and 1864 survived for up to 10 years after hospitalization. Although this cohort may have included cases of ENL, these findings more likely reflect the generally poor living and health conditions of the period. We believe that these contextual factors should be taken into account and discussed when interpreting these data.

Lines 508-509: It is clear that the manuscript identifies weaknesses in all current classifications of ENL. However, the authors’ proposed improvements are not clearly articulated, aside from their suggestion to conduct a Delphi study involving experts.

Reviewer #3: No revisions suggested.

**Summary and General Comments**

Reviewer #1: (No Response)

Reviewer #2: This manuscript addresses a highly relevant and timely topic in leprology. The lack of consensus and clarity surrounding the temporal classification of erythema nodosum leprosum (ENL) has important implications for clinical practice, research comparability, and therapeutic decision-making. By combining contemporary literature with descriptions from the pre-corticosteroid and pre-antibiotic era, the authors provide a valuable historical perspective that enriches current discussions and highlights fundamental inconsistencies in existing classifications.

The review is comprehensive and thoughtfully constructed, and its core premise—that current ENL classifications suffer from conceptual and practical limitations—is well supported by the material presented. With some clarification of the methodological approach, improved distinction between the narrative and systematic components, and further development of the Discussion to more explicitly articulate the implications of the identified overlaps between acute, recurrent, and chronic ENL, this manuscript has the potential to make a significant contribution to the field.

With the aim of contributing constructively to the manuscript, I have suggested a limited number of adjustments, mainly related to clarifying the methodological design and improving the distinction between the methods used for the narrative and systematic review approaches. In addition, we believe that the conceptual overlap among ENL classifications should be more explicitly acknowledged and expanded upon in the Discussion, as this represents a central conclusion of the study.

We have also recommended greater caution when aligning historical descriptions with contemporary classifications, for example, when accepting the interpretation that former nodular forms can be translated as lepromatous leprosy, and when attributing the short survival of patients in the first half of the nineteenth century primarily to ENL.

Expanding the Discussion would further strengthen the manuscript’s contribution and provide clearer guidance for future efforts to refine ENL classification.

Overall, the recommendations relate to relatively minor points and are intended to support the further improvement of an otherwise well-conducted and well-written study. We hope that the specific suggestions will be helpful to the authors.

Reviewer #3: The article explores an important aspect of ENL. Since ENL leads to high morbidity and contributes significantly to the chronicity of leprosy, uniform descriptions and widely accepted definitions and classifications are necessary to ensure the comprising of research carried out in diverse aspects in leprosy and ENL.

The current work looks at the weaknesses of the present classification system and recommends how best to overcome these weaknesses.

PLOS authors have the option to publish the peer review history of their article (what does this mean? ). If published, this will include your full peer review and any attached files.). If published, this will include your full peer review and any attached files.

**Do you want your identity to be public for this peer review?** For information about this choice, including consent withdrawal, please see our For information about this choice, including consent withdrawal, please see our Privacy Policy ..

Reviewer #1: No

Reviewer #2: **Yes:** Mauricio Lisboa Nobre, MD, PhDMauricio Lisboa Nobre, MD, PhD

Reviewer #3: No

**Figure resubmission:**While revising your submission, we strongly recommend that you use PLOS’s NAAS tool (https://ngplosjournals.pagemajik.ai/artanalysis) to test your figure files. NAAS can convert your figure files to the TIFF file type and meet basic requirements (such as print size, resolution), or provide you with a report on issues that do not meet our requirements and that NAAS cannot fix.While revising your submission, we strongly recommend that you use PLOS’s NAAS tool (https://ngplosjournals.pagemajik.ai/artanalysis) to test your figure files. NAAS can convert your figure files to the TIFF file type and meet basic requirements (such as print size, resolution), or provide you with a report on issues that do not meet our requirements and that NAAS cannot fix.
---

## [Decision Letter · Decision Letter 1]

27 Feb 2026

Dear Dr. de Barros,

We are pleased to inform you that your manuscript 'Descriptions of the natural history of erythema nodosum leprosum to inform clinical classification – a semi-systematic review' has been provisionally accepted for publication in PLOS Neglected Tropical Diseases.

Best regards,

Elsio A Wunder Jr, DVM, Ph.D.

Section Editor

Elsio Wunder Jr

Section Editor

Shaden Kamhawi

co-Editor-in-Chief

Paul Brindley

co-Editor-in-Chief

Reviewer's Responses to Questions

**Key Review Criteria Required for Acceptance?**

**Methods**

-Are the objectives of the study clearly articulated with a clear testable hypothesis stated?

-Is the study design appropriate to address the stated objectives?

-Is the population clearly described and appropriate for the hypothesis being tested?

-Is the sample size sufficient to ensure adequate power to address the hypothesis being tested?

-Were correct statistical analysis used to support conclusions?

-Are there concerns about ethical or regulatory requirements being met?

Reviewer #2: (No Response)

**Results**

-Does the analysis presented match the analysis plan?

-Are the results clearly and completely presented?

-Are the figures (Tables, Images) of sufficient quality for clarity?

Reviewer #2: (No Response)

**Conclusions**

-Are the conclusions supported by the data presented?

-Are the limitations of analysis clearly described?

-Do the authors discuss how these data can be helpful to advance our understanding of the topic under study?

-Is public health relevance addressed?

Reviewer #2: (No Response)

**Editorial and Data Presentation Modifications?**

Reviewer #2: (No Response)

**Summary and General Comments**

Reviewer #2: Dear Editor,

Thank you for the opportunity to review once again the manuscript submitted by Dr. de Barros et al., entitled “Descriptions of the Natural History of Erythema Nodosum Leprosum to Inform Clinical Classification – A Semi-Systematic Review,” following the authors’ revisions.

The authors have made the necessary adjustments to several of the points previously raised, particularly with the aim of improving the clarity and overall readability of the text. At the same time, they addressed instances in which their interpretation differed from ours, appropriately presenting and substantiating their perspectives with arguments that are both relevant and valuable to the manuscript.

Accordingly, we have no further comments to add and recommend the publication of this important article for readers seeking a better understanding of the clinical course of leprosy, especially with regard to the management of erythema nodosum leprosum.

Sincerely,

Mauricio Lisboa Nobre

PLOS authors have the option to publish the peer review history of their article (what does this mean? ). If published, this will include your full peer review and any attached files.). If published, this will include your full peer review and any attached files.

**Do you want your identity to be public for this peer review?** For information about this choice, including consent withdrawal, please see our For information about this choice, including consent withdrawal, please see our Privacy Policy ..

Reviewer #2: No

---

## [Editor Report · Acceptance letter]

Dear Dr. de Barros,

We are delighted to inform you that your manuscript, "Descriptions of the natural history of erythema nodosum leprosum to inform clinical classification – a semi-systematic review," has been formally accepted for publication in PLOS Neglected Tropical Diseases.

Best regards,

Shaden Kamhawi

co-Editor-in-Chief

Paul Brindley

co-Editor-in-Chief
